# Routine Brain MRI Findings on the Long-Term Effects of COVID-19: A Scoping Review

**DOI:** 10.3390/diagnostics13152533

**Published:** 2023-07-30

**Authors:** Yuriy Vasilev, Ivan Blokhin, Anna Khoruzhaya, Maria Kodenko, Vasiliy Kolyshenkov, Olga Nanova, Yuliya Shumskaya, Olga Omelyanskaya, Anton Vladzymyrskyy, Roman Reshetnikov

**Affiliations:** 1Research and Practical Clinical Center for Diagnostics and Telemedicine Technologies, Department of Health Care of Moscow, Russian Federation, Petrovka Street, 24, Building 1, 127051 Moscow, Russia; 2Department of Biomedical Technologies, Bauman Moscow State Technical University, 2nd Baumanskaya Street, 5, Building 1, 105005 Moscow, Russia; 3Department of Information and Internet Technologies, I.M. Sechenov First Moscow State Medical University, Trubetskaya Street, 8, Building 2, 119991 Moscow, Russia

**Keywords:** long COVID, imaging, magnetic resonance, diagnostic tests, routine, scoping review

## Abstract

Rationale and Objectives: Post-COVID condition (PCC) is associated with long-term neuropsychiatric symptoms. Magnetic resonance imaging (MRI) in PCC examines the brain metabolism, connectivity, and morphometry. Such techniques are not easily available in routine practice. We conducted a scoping review to determine what is known about the routine MRI findings in PCC patients. Materials and Methods: The PubMed database was searched up to 11 April 2023. We included cohort, cross-sectional, and before–after studies in English. Articles with only advanced MRI sequences (DTI, fMRI, VBM, PWI, ASL), preprints, and case reports were excluded. The National Heart, Lung, and Blood Institute and PRISMA Extension tools were used for quality assurance. Results: A total of 7 citations out of 167 were included. The total sample size was 451 patients (average age 51 ± 8 years; 67% female). Five studies followed a single recovering cohort, while two studies compared findings between two severity groups. The MRI findings were perivascular spaces (47%), microbleeds (27%) and white matter lesions (10%). All the studies agreed that PCC manifestations are not associated with specific MRI findings. Conclusion: The results of the included studies are heterogeneous due to the low agreement on the types of MRI abnormalities in PCC. Our findings indicate that the routine brain MRI protocol has little value for long COVID diagnostics.

## 1. Introduction

While COVID-19 is slowly fading from the headlines [1], a growing number of people [2,3] report persistent symptoms of the condition that has become internationally recognized as long COVID or post-COVID condition (PCC) [4]. The disease is associated with over 60 heterogeneous physical and psychological symptoms affecting multiple organ systems [5,6]. Such heterogeneity generates sometimes controversial and confusing findings. For example, according to the Centers for Disease Control-funded INSPIRE (Innovative Support for Patients with SARS-CoV-2 Infections Registry) group’s results, COVID-negative patients reported similar or higher rates of symptoms usually associated with PCC, such as fatigue, fever, headache, runny nose, sore throat, and poor well-being [7,8]. These controversies also extend to medical imaging research. In a large longitudinal brain imaging study, Douaud et al. identified significant effects of SARS-CoV-2 infection, including reduction in grey matter thickness and global brain size [9]. In contrast, Yiping et al. reported a higher bilateral grey matter volume and no significant changes in the white matter volume in COVID-positive patients [10]. In addition to conflicting results, these studies used advanced neuroimaging techniques instead of the routine brain screen MRI protocol, which is likely to be employed in case of non-specific symptoms [11]. The neuroimaging PCC signs detected using diffusion tensor imaging, the total and regional brain volume, in the aforementioned citations are difficult to incorporate into existing diagnostic pathways. The overall protocol may take twice as long [12,13] or may need advanced image reconstruction techniques to shorten its duration [14]. The study reporting time would also be prolonged by computationally intensive post-processing. The decision making on the widespread use of advanced MRI techniques should consider, among others, non-health-related factors such as healthcare resource utilization [15]. Provided that the implementation of broad, comprehensive, and low-cost protocols is the priority for medical institutions [16], the capabilities of a routine brain MRI for long COVID imaging should be carefully studied.

The information available in the literature about the neurological sequelae of COVID-19 provides conflicting, confusing, and heterogeneous findings. The existing knowledge regarding imaging findings is not always applicable to routine clinical practice; moreover, visual MRI abnormalities in PCC patients have not been systematically searched and summarized. For these reasons, a scoping review is essential to map the research performed in this area and identify gaps in knowledge. The following research question was formulated: What is known from the published literature about the changes in the brain observed on routine MRI in adult patients who have recovered from the symptoms of acute COVID-19?

## 2. Materials and Methods

### 2.1. Study Design

In this scoping review, we mapped the data published from inception up to 11 April 2023 (the date of the most recent search) for studies that report the brain MRI results of people with post-COVID conditions. We searched PubMed for peer-reviewed publications using the query (“Post-Acute COVID-19 Syndrome”(mh) OR post-acute COVID OR long haul COVID OR long COVID OR post COVID) AND (“Diagnostic imaging”(mh) OR “Magnetic Resonance Imaging”(mh) OR magnetic resonance imaging OR MRI OR MRI scan) AND (head(mh) OR head OR brain(mh) OR brain) AND (findings OR abnormalities OR entities OR lesions OR sequalae) NOT (“Case Reports”(pt) OR “Review”(pt)). The search was limited to the English language. The final search results were exported into Mendeley Reference Manager and checked for duplicates using built-in tools. This study complied with the PRISMA-ScR (Preferred Reporting Items for Systematic reviews and Meta-Analyses extension for Scoping Reviews) guidelines [17].

### 2.2. Inclusion and Exclusion Criteria

Any study with the results of routine brain MRI of adult (18+ years old) patients who recovered from the symptoms of acute COVID-19, regardless of the presence of post-COVID-associated health problems, was included in our analysis. We included retrospective and prospective cohort studies, cross-sectional studies, and before–after studies with no control group. We excluded clinical cases, case series, letters to editors, preprints, pediatric studies, and literature reviews. We defined routine brain MRI as at least a four-sequence MRI protocol that consists of T1-weighted images, T2-weighted fluid-attenuated inversion recovery (FLAIR) images, diffusion-weighted images (DWI) with apparent diffusion coefficient (ADC) maps, and T2*-weighted images, in accordance with Mehan et al. [18]. We also included contrast-enhanced MRI studies, studies with isotropic (three-dimensional) sequences (i.e., CUBE, VISTA, SPACE), and studies that used susceptibility-weighted imaging (SWI). We excluded from the analysis studies that employed any of the following techniques: diffusion tensor imaging, functional MRI, voxel-based brain morphometry, and perfusion-weighted imaging, including arterial spin labeling. If no details were provided by the publication authors on the MRI image acquisition, we assumed that it was performed using the routine protocol and included the citation. Additionally, we included studies that used a technique from the exclusion criteria list but also presented the results of routine brain MRI. Publications on mucormycosis or other comorbidities associated with COVID-19 were excluded from the analysis regardless of the MRI protocol used because of possible confounding due to the co-infection-related findings.

### 2.3. Study Selection and Data Extraction

We used a two-stage approach to screen citations for inclusion. In the first stage, two independent reviewers screened the titles and abstracts against the inclusion criteria. The second step, also performed by two independent screeners, was to evaluate the main text and supplementary materials of the publications that passed the first step. To minimize selection bias and human error, we performed screener training and pilot testing of the screening criteria on a sample of 14 studies. The screening and selecting process was documented in real life, including providing the reasons for article exclusion.

We extracted the data on all the abnormalities indicated by routine brain MRI and their localization, if provided. Where available, we extracted the statistical test results concerning the significance of the brain changes in size or quantity between the before and after time periods, or between the exposed and control groups. When these data were not available, we extracted descriptive statistics on the type and incidence of a particular finding.

The complete information extracted included the author, article information (publication date, journal, impact factor), location, study objective, design, type (multicenter or single center), and inclusion and exclusion criteria, study population, including COVID-19 severity distribution, follow-up time, and study results (Table 1 and Table 2).

The data extraction was performed by one reviewer and independently verified by a second reviewer. In case of disagreements between the reviewers in terms of study selection and data extraction decisions, a third reviewer was consulted.

### 2.4. Critical Appraisal of Sources of Evidence

Each included citation was evaluated independently by two reviewers using the National Heart, Lung, and Blood Institute (NHLBI) tools according to the study design [26]. In case of disagreements between reviewers in terms of study evaluation decisions, a third reviewer was consulted. The NHLBI tools consist of a series of questions, which we have tailored in regard of the properties of the included studies. For the observational cohort studies and cross-sectional studies, we omitted the questions on:

(1) Blinding the radiologists assessing the MRI results to the patients’ exposures, since in this case blinding to the clinical data may complicate the image interpretation. It is important to consider clinical and functional data of the patient that can affect the interpretation results, especially in case of co-infections.

(2) Multiple measurements of the outcome of interest. The research question addresses primarily the presence and type of the brain abnormalities, which do not require repeated measurements on the same MRI scan.

The NHLBI tool guidelines recommend assigning “no” as an answer to the question on the sufficiency of the time frame between exposure and overcome for cross-sectional designs. Within this study, we defined exposure as laboratory-confirmed COVID-19 with three severity categories: (i) mild (non-hospitalized); (ii) hospitalized, non-ICU; and (iii) hospitalized, admitted to ICU. All the included studies had a sufficient time window for the exposure to have an effect on the outcome (PCC with visually detectable brain MRI abnormalities). Thus, we assigned “yes” as an answer to the time frame question for cross-sectional studies too.

The final rating (good, fair or poor) was assigned based on the proportion of “yes” answers: poor for <50%, fair for <80%, and good for ≥80%. The answer “NA” did not count negatively toward the quality rating. All the studies were treated equally regardless of the quality rating.

### 2.5. Synthesis of Results

We collected the data on the types of abnormalities reported in the included studies and summarized the MRI protocol details, populations and study designs.

## 3. Results

The search of the PubMed database identified 167 citations. Seven studies met our inclusion criteria and were included into this review (Figure 1).

All of the studies originated from different high-income countries, namely the Netherlands [19], Japan [20], Israel [21], France [22], Italy [23], the USA [24], and Sweden [25]. The authors of only three studies reported receiving research grants.

The total population of the 7 included studies was 451 patients, with an average age of 51 ± 8 years old. A total of 2 studies recruited patients older than 15 years of age, but we decided to not exclude them because the mean and median age in these citations were 40 and 59 years old, respectively [20,25]. All seven studies had participants of any gender/sex, with the majority of the patients being female/women (67%). None of the studies commented on the inclusion of diverse participants with nonbinary gender identities. The authors of two citations provided the racial/ethnic characteristics of the study population, with one study performed among Asian patients [20] and another study recruiting Jewish/Arab participants [21].

According to the results of a critical appraisal of the included citations, three studies had a good-quality rating, one study had a fair-quality rating, and three studies had a poor-quality rating (Table 3).

The minimum time period elapsed since the COVID-19 diagnosis or hospital discharge within the sample was 2 months [20], with an average follow-up period of 220 days and a longest-lasting self-reported symptom duration of 462 days. Patients from five studies had COVID-19 verified by polymerase chain reaction test or antigen test during the acute phase of the disease. The authors of two studies claimed that they recruited patients with confirmed COVID-19 but did not specify the confirmation method, inducing uncertainty as to the true SARS-CoV-2 infection status of the participants [19,21]. For example, the study by Klinkhammer et al. included patients who were admitted to one of the six recruiting Dutch hospitals during the first European infection wave [19], when there was a great shortage of diagnostic test materials [27,28].

The study designs used by the authors included prospective cohort, retrospective cohort, and cross-sectional. Of the seven studies, five followed a single cohort of patients recovered from acute COVID-19. Two studies compared the COVID-19 sequelae between two groups: hospitalized versus non-hospitalized [24], and ICU versus non-ICU patients [19]. None of the studies included a control group of COVID-negative patients for brain MRI findings.

### 3.1. Image Acquisition Protocols

Three studies did not report the details of the brain MRI protocol. The other four used a 3-Tesla MR scanner with the following sequences: T1- and T2-weighted, fluid-attenuated inversion recovery, and susceptibility- and diffusion-weighted imaging. For four patients in one study, scans acquired using a 1.5-Tesla MR system with T2* instead of SWI were used. One study used MR angiography in some cases in addition to the standardized protocol, and one study added 3D T1-gradient echo imaging to the routine technique.

### 3.2. Brain MRI Findings

All the included studies considered brain lesions as possible consequences of COVID-19, without explicitly addressing other risk factors. Data on the imaging results were not sufficient to conduct a meta-analysis. The most common finding in patients with long-lasting symptoms was perivascular spaces, observed in 47% of the total study population (Table 4). Note that in one citation the incidence of perivascular spaces was close to 100%, whereas the majority of included studies did not report this finding (Table 4). Most of the lesion types were unique, not reproducing in other papers. One study reported the complete absence of abnormalities on the brain MRI scans. The studies by Klinkhammer et al. and Hellgren et al. had the greatest number of overlapping findings, namely perivascular spaces, microbleeds, white matter lesions, and global cortical atrophy (Table 4).

White matter lesions were the only abnormality reported in the majority of the included studies (six out of seven, Table 3). In the study by Hellgren et al., six patients had baseline MRI scans obtained in the acute phase during their hospitalization. All six of them had a higher number of white matter lesions at the follow-up scan performed about seven months after admittance to the hospital. Note that when considering the cross-sectional data, there was no direct link between the condition and the finding; only 40/445 (9%) PCC patients with single-point MRI results had white matter lesions (Table 4). The clinical significance of these findings is not clear. According to Bungerberg et al., there was no association between the severity of the white matter lesions and the clinical outcomes. A similar statement holds for the other findings. All the studies but one agreed that the neurological symptoms reported by the patients were not related to brain MRI lesions. The only exception is the work of Hellgren et al., which showed that the patients with MRI abnormalities had a lower visuospatial index than the patients with normal MRI. However, the patients from the first group were significantly older (median age 62.0 versus 50.5 years, *p* = 0.007). Since aging is associated with a decline in visual processing [29], this age-related decline could confound the analysis, being partially responsible for the results obtained by the authors.

The work by Hellgren et al. was the only study that performed a before–after comparison for some of the findings. The other six provided a single-point characterization of the study population, without assessing the imaging data of the patients before the SARS-CoV-2 infection or during the acute phase of COVID-19.

## 4. Discussion

In this scoping review, we identified seven studies addressing PCC biomarkers that can be detected on routine brain MRI, which were published until April 2023. Our findings indicate that the neurophysiological manifestations after COVID-19 infection are not associated with the presence of visually detectable brain MRI lesions. There was one case of a statistically significant relation between abnormal MRI (white matter lesions) and neurophysiological malfunctioning. However, the authors did not control their findings for obvious confounding factors such as age; moreover, the sample size was small, with only four patients below the cut-off demarcating impairment [25]. We consider this result as inconclusive and not influencing the main pool of evidence suggesting that PCC-associated symptoms are not related to white matter lesions on MRI. Despite that, the longitudinal data of Hellgren et al. indicate that there may be a possible correlation between this abnormality type and long COVID diagnosis, which requires further research.

There was a slight contradiction between the studies of Klinkhammer et al. and Bungerberg et al. regarding the association between microbleeds and the severity of COVID-19. In the work by Klinkhammer et al., microbleeds were observed both in the ICU and non-ICU groups, with the ICU patients having microbleeds more often than the non-ICU patients (61% versus 32%, *p* < 0.001). Bungerberg et al. discovered microbleeds almost exclusively in the ICU patients who required extracorporeal membrane oxygenation support, with the only exception being one patient with known amyloidosis. However, there are two factors that cast uncertainty on the direct link between COVID-19 and cerebral microbleeds. First, there is an established association between ICU care and microvascular pathology incidence, both for COVID-19 and non-COVID-19 patients [30,31]. Second, Klinkhammer et al. performed a linear regression analysis with the number of microbleeds as a predictor of cognitive disfunction in PCC patients, which has not shown any significant relationship (β  =  0.31, *p*  =  0.80). Thus, although there is a possibility of perceiving microbleed occurrence in non-ICU COVID-19 patients, they do not seem to contribute to long COVID presentation. The number of cerebral microbleeds tends to increase during normal aging [32], and the median age in the non-ICU group of the Klinkhammer et al. study was 64 years. Moreover, according to Barnaure et al., there is no clear correlation between microbleeds and cognitive symptoms in healthy controls [33]. Thus, the available data provide limited evidence of a relationship between microbleeds and PCC.

The types of findings reported in the included studies are heterogeneous, inconsistent and not always clearly defined. For example, Hellgren et al. reported that a fraction of patients in their study had susceptibility-weighted image abnormalities with no further explanation. SWI is commonly used to discriminate microbleeds from microcalcifications, although proper interpretation is not always possible and sometimes requires additional testing [34]. It is a potentially important distinction for PCC; while microbleeds do not seem to contribute to cognitive malfunctioning, there are case reports attributing brain calcifications in COVID-19 survivors to neurological symptoms [35,36]. However, the data on the prevalence and clinical significance of microcalcifications in long COVID are scarce and require more in-depth research.

The most common brain MRI abnormality identified in this review was perivascular spaces. In the study by Klinkhammer et al., they were observed in 187 out of 188 patients. However, the other included studies, with the exception of Bungerberg et al., did not report this finding. This might be due to the fact that perivascular spaces are considered a normal finding and their function is important for maintaining brain health [37]. Bungerberg et al. focused specifically on enlarged perivascular spaces (EPVS), since in this case dilation is a marker of dysfunction [37]. This may explain the difference in the reported incidence between the two studies (26 out of 50 versus 187 out of 188, Table 3). Nevertheless, according to Bungerberg et al., there were no associations between EPVS and clinical outcomes in the PCC patients.

Existing evidence indicates that the applicability of medical imaging, including molecular imaging, remains ambiguous for the diagnosis and management of PCC [38]. According to CDC information for healthcare providers, imaging study findings can often be normal or nondiagnostic in patients experiencing post-COVID conditions [39]. This information is consistent with our findings, which demonstrate that none of the 12 types of reported brain MRI abnormalities has any significant correlation with long COVID symptoms. The results of this review suggest that the employment of routine brain MRI for PCC patients should be reserved for potentially life-threatening conditions, when necessary.

This scoping review has some limitations. We searched a single database, excluded preprints, and did not include grey literature. Because of this, we could have missed some potentially relevant results. Due to the heterogeneity of the types of brain MRI abnormalities, we did not conduct a meta-analysis. However, despite the low agreement between the studies on the types of findings, all the included citations were consistent on the lack of clinical significance of these abnormalities.

## Figures and Tables

**Figure 1 diagnostics-13-02533-f001:**
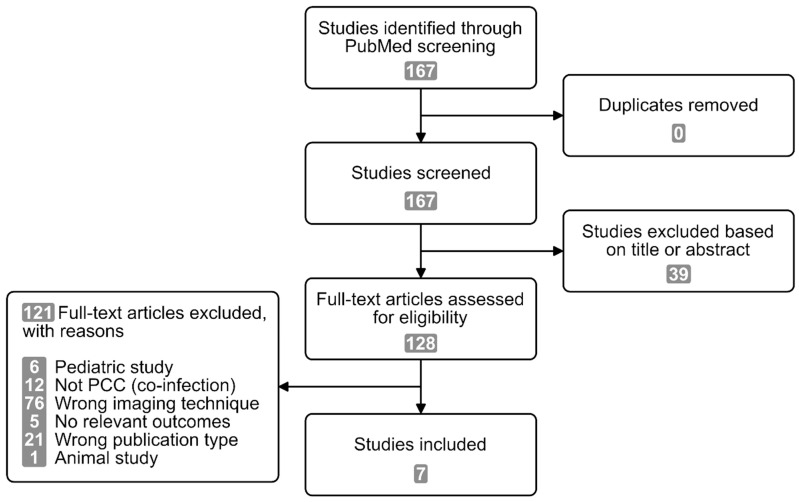
PRISMA diagram of the selection of sources of evidence.

**Table 1 diagnostics-13-02533-t001:** The studied literature.

First Author, Year	Title	Journal, Impact Factor	Location	Objective	Datapoints for MRI Imaging	Design
Klinkhammer, 2023 [19]	Neurological and (neuro)psychological sequelae in intensive care and general ward COVID-19 survivors	*European Journal of Neurology*, 6.3	The Netherlands	To investigate whether COVID-19 ICU-admitted patients are more prone to brain abnormalities and neurological and (neuro)psychological consequences than non-ICU patients	Single, post-COVID	Prospective cohort, multicenter
Ohira, 2022 [20]	Clinical features of patients who visited the outpatient clinic for long COVID in Japan	*eNeurologicalSci*, 0.6	Japan	To examine the clinical characteristics of patients with long COVID in Japan	Single, post-COVID	Retrospective cohort, single center
Hadad, 2022 [21]	Cognitive dysfunction following COVID-19 infection	*Journal of NeuroVirology*, 3.7	Israel	To improve the characterization of the cognitive impairment of patients recovering from COVID-19 infection	Single, post-COVID	Prospective cohort, single center
Kachaner, 2022 [22]	Somatic symptom disorder in patients with post-COVID-19 neurological symptoms: a preliminary report from the somatic study (Somatic Symptom Disorder Triggered by COVID-19)	*Journal of Neurology, Neurosurgery and Psychiatry*, 13.6	France	To determine whether a positive diagnosis of SSD can be asserted in patients with long-lasting neurological symptoms occurring after mild COVID-19	Single, post-COVID	Prospective cohort, single center
Taruffi, 2022 [23]	Neurological Manifestations of Long COVID: A Single-Center One-Year Experience	*Neuropsychiatric Disease and Treatment*, 3.0	Italy	To report a single-center experience of the neurological manifestations of long COVID	Single, post-COVID	Cross-sectional, single center
Bungerberg, 2022 [24]	Long COVID-19: Objectifying most self-reported neurological symptoms	*Annals of Clinical and Translational Neurology*, 5.4	The USA	To objectify and compare persisting self-reported symptoms in initially hospitalized and non-hospitalized patients after infection with severe acute respiratory syndrome	Single, post-COVID	Cross-sectional, single center
Hellgren, 2021 [25]	Brain MRI and neuropsychological findings at long-term follow-up after COVID-19 hospitalisation: an observational cohort study	*BMJ Open*, 3.0	Sweden	To report the association between brain MRI findings and neurocognitive function, as well as persisting fatigue at long-term follow-up after COVID-19 hospitalization in patients identified as at high risk of CNS affection	For 6/35 patients: two, acute phase and post-COVID	Prospective cohort, single center

**Table 2 diagnostics-13-02533-t002:** The studied literature (additional data).

First Author, Year	Inclusion Criteria	Exclusion Criteria	Demographics: Mean or Median Age, Total Population, Female Sex, Patients with MRI	COVID-19 Severity Distribution during Acute Phase	Median Follow-Up Duration, Days	Study Findings
Klinkhammer, 2023 [19]	Patients ≥ 18 y.o. admitted to one of the recruiting hospitals from March to June 2020 for the treatment of COVID-19, at least six months post hospital discharge	Individuals with MRI contra-indications, cognitive impairment prior to hospital admission, physical inability to visit a hospital, or new severe neurological damage after hospital discharge	64 and 61 years (non-ICU and ICU group, respectively) 205 patients total 61 females 188 MRIs	104 hospitalized, non-ICU 101 hospitalized, ICU	244	No significant relationship between brain abnormalities and cognitive dysfunction (β = 0.31, *p* = 0.80)
Ohira, 2022 [20]	Patients ≥ 15 y.o. admitted to the hospital from 1 June to 31 December 2021 reporting PCC symptoms, at least two months since the diagnosis of COVID-19 or the end of hospitalization	None	39.8 years 90 patients total 51 females 42 MRIs	50 non-hospitalized 36 hospitalized, no data on ICU admittance 4 patients, no data	122	Four patients had sinusitis, three of them exhibited smell/taste disturbance; however, the link between MRI findings and patients’ symptoms is unclear
Hadad, 2022 [21]	Patients attending post-COVID clinic from December 2020 to June 2021, with cognitive symptoms, at least six weeks after infection	None	50 years 46 patients total 30 females No data on quantity of patients with MRI	31 non-hospitalized 15 hospitalized, no data on ICU admittance	183	MRI images did not reveal alternative etiologies for the cognitive syndrome
Kachaner, 2022 [22]	All adult consecutive patients referred to the hospital for post-COVID consultation from May 2020 to April 2021	Patients hospitalized during the acute phase and those with suspected de novo neurological pathology unrelated to COVID-19	46 years 50 patients total 41 females 49 MRIs	50 non-hospitalized	425	The rate of MRI abnormalities was in accordance with the general population, arguing for non-specific findings
Taruffi, 2022 [23]	Patients attending the “long NeuroCOVID” clinic from 21 January to 9 December 2021, with a persistent neurological disturbance, at least one month after acute COVID-19 or its resolution	None	50.5 years 103 patients total 62 females 41 MRIs	79 non-hospitalized 21 hospitalized, non-ICU 3 hospitalized, ICU	243	MRI did not show pathological findings in the vast majority of patients
Bungerberg, 2022 [24]	Patients ≥ 18 y.o. recruited from different departments of the hospital from 13 August 2020 to 30 March 2021, with persisting symptoms for at least four weeks	None	50.5 years 50 patients total 28 females 42 MRIs	29 non-hospitalized 10 hospitalized, non-ICU 11 hospitalized, ICU	205	No association was found between MRI findings and clinical outcomes, with the exception of cerebral microbleeds almost exclusively found in hospitalized patients who received extracorporeal membrane oxygenation support
Hellgren, 2021 [25]	Patients ≥ 15 y.o. who were admitted to the hospital from 1 March to 31 May 2020 for treatment of COVID-19, reporting PCC symptoms at four months after discharge	Patients with severe comorbidities, non-COVID patients, patients without PCC symptoms or without concerning findings	59 years 35 patients total 7 females 35 MRIs	15 hospitalized, non-ICU 20 hospitalized, ICU	122	The visuospatial index value was lower in the group with abnormal MRI compared with the group with normal MRI (mean 81.8 vs. 94.3, *p* = 0.031). Otherwise, there were no between-group differences regarding neurocognition, fatigue, depression or anxiety

**Table 3 diagnostics-13-02533-t003:** Critical appraisal of the sources of evidence.

First Author, Year	Clear Objective	Defined Population	Participation Rate ≥ 50%	Inclusion/Exclusion Criteria Prespecified	Sample Size Justified	Exposures Measured Prior to Outcomes	Sufficient Timeframe	Levels of Exposure Examined	Exposure Measures Defined	Outcome Measures Defined	Loss to Follow-Up <20%	Confounding Variables Measured	Rating
Klinkhammer, 2023 [19]	Yes	Yes	No	Yes	Yes	Yes	Yes	NA	Yes	Yes	Yes	Yes	Good
Ohira, 2022 [20]	Yes	Yes	Yes	Yes	No	No	Yes	No	CD	No	NA	No	Poor
Hadad, 2022 [21]	Yes	No	No	No	No	Yes	Yes	NA	Yes	No	NA	No	Poor
Kachaner, 2022 [22]	Yes	Yes	Yes	Yes	No	Yes	Yes	Yes	Yes	No	Yes	Yes	Good
Taruffi, 2022 [23]	CD	Yes	Yes	CD	No	Yes	Yes	Yes	Yes	No	NA	No	Poor
Bungerberg, 2022 [24]	Yes	Yes	Yes	Yes	No	Yes	Yes	Yes	Yes	Yes	NA	Yes	Good
Hellgren, 2021 [25]	Yes	Yes	No	Yes	No	Yes	Yes	Yes	Yes	No	NA	No	Fair

**Table 4 diagnostics-13-02533-t004:** Number of PCC patients with brain MRI abnormalities.

First Author, Year	Population/Findings Number **	Findings, Count (%) *											
		Perivascular Spaces	Microbleeds	White Matter Lesions	Lacunes	Global Cortical Atrophy	Cerebral Infarcts	Macrobleeds	SWI Abnormalities	Medial Temporal Lobe Atrophy	Sinusitis	Mild Cortical Atrophy	Venous Angioma
Klinkhammer, 2023 [19]	188/331	187 (40.6)	92 (19.9)	2 (0.4)	25 (5.4)	2 (0.4)	11 (2.4)	8 (1.7)		4 (0.9)			
Ohira, 2022 [20]	42/6			1 (0.2)							4 (0.9)		1 (0.2)
Hadad, 2022 [21]	46/0												
Kachaner, 2022 [22]	49/8			8 (1.7)									
Taruffi, 2022 [23]	41/4			2 (0.4)								2 (0.4)	
Bungerberg, 2022 [24]	50/79	26 (5.6)	29 (6.3)	8 (1.7)		16 (3.6)							
Hellgren, 2021 [25]	35/33			25 (5.4)					8 (1.7)				
Total (%)	451/461	213 (46.2)	121 (26.2)	46 (10.0)	25 (5.4)	18 (4.0)	11 (2.4)	8 (1.7)	8 (1.7)	4 (0.9)	4 (0.9)	2 (0.4)	1 (0.2)

* Percentage of the finding in an individual study among the total number of findings is shown. ** Only patients with brain MRI data are included from individual studies.

## Data Availability

Data sharing not applicable.

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
