# Peer review of "Routine Brain MRI Findings on the Long-Term Effects of COVID-19: A Scoping Review"

_diagnostics, 2023, doi:10.3390/diagnostics13152533_

Round 1

Reviewer 1 Report

This review paper summarized the findings of MRI abnormalities in patients with Post-COVID condition (PCC). This investigation is very important for the outcome of patients with long COVID. However, It does not contain appropriate data for the journal of Diagnostics.

1. This paper reviewed the differenct studies in datapoints for MRI imaging of single and two points. It is difficult to interpret these data in a unified way and is likely to cause confusion for the reader.

2. Author stated in the second paragraph of the discussion, "We believe that no further research is needed to establish a possible relationship between microbleeds and PCC due to its low probability and lack of clinical value." It would be an overstatement to say that there is no need for further consideration, because of lack of the evidence for meta-analysis.

3. Table1 has too many items and is difficult to read. It would be better to separate the Tables.

4. In Table 3, the format of Klinkhammer, 2023 is different from the other studies. (e.g. only Klinkhammer, 2023 is underlined)

The descriptions in English are generally unproblematic.

Author Response

Responses to Reviewer 1 comments on the manuscript entitled “Routine Brain MRI Findings on Long-Term Effects of COVID-19: A Scoping Review”

This review paper summarized the findings of MRI abnormalities in patients with Post-COVID condition (PCC). This investigation is very important for the outcome of patients with long COVID. However, It does not contain appropriate data for the journal of Diagnostics.

We thank the referee for acknowledging the importance of the research question of our review. We cannot agree with the reviewer’s opinion regarding the inappropriateness of the manuscript for the Diagnostics journal. Radiology is in the journal’s scope (https://www.mdpi.com/journal/diagnostics/about); moreover, we submitted the manuscript to the special issue titled “Diagnosis and Management in COVID-19 Patient” (https://www.mdpi.com/journal/diagnostics/special_issues/09HBJ3FC4Z), addressing radiology biomarkers in patients with post-COVID condition.

  1. This paper reviewed the differenct studies in datapoints for MRI imaging of single and two points. It is difficult to interpret these data in a unified way and is likely to cause confusion for the reader.

Our review includes seven studies, six of which contain a cross-sectional, single-point evaluation of brain changes in PCC patients (total population 451). The remaining study contained two datapoints (acute phase and post-COVID) only for 6 out of 35 patients (Table 1). All of these six patients had additional white matter lesions at the follow-up MRI. Multiple white matter lesions were the main MRI finding in the study population, including the patients with a single data point (Hellgren, 2021). Therefore, two datapoints were available only for ~1% of the review population (Table 1), and their results were consistent with the main body of evidence (Table 4).

In response to the comment, we have revised the text of the section 3.2 and Discussion. We hope that the manuscript text is now improved and easier to follow.

  1. Author stated in the second paragraph of the discussion, "We believe that no further research is needed to establish a possible relationship between microbleeds and PCC due to its low probability and lack of clinical value." It would be an overstatement to say that there is no need for further consideration, because of lack of the evidence for meta-analysis.

In the revised version of the manuscript, we have removed the mentioned statement, replacing it with an assertion on the limited evidence for a relationship between microbleeds and PCC (Discussion, second paragraph).

  1. Table1 has too many items and is difficult to read. It would be better to separate the Tables.

We apologize for the inconvenience. In the revised version of the manuscript, Table 1 was split into two separate tables.

  1. In Table 3, the format of Klinkhammer, 2023 is different from the other studies. (e.g. only Klinkhammer, 2023 is underlined)

We thank the reviewer, and Table 3 was fixed.

Reviewer 2 Report

Overall nice review. The aim of this review article is to focus on brain changes in Covid19 patients. I agree with the authors that there are limited studies available about brain tissue changes in covid 19 patients, however, a recent search showed that there are at least 4-5 data-based articles that showed the grey matter changes in patients with Covid-19, I would suggest to authors to includes those studies in the review article as the main focus of this article is brain changes and include some figures from those studies if available. Voxel-based brain morphometry: this is the post-processing method and usually runs on the T1 weighted imaging, I think this should include in the analysis as you include the T1 weighted higher resolution imaging to look for grey matter changes.

Author Response

Response to Reviewer 2 comments on the manuscript entitled “Routine Brain MRI Findings on Long-Term Effects of COVID-19: A Scoping Review”

Overall nice review. The aim of this review article is to focus on brain changes in Covid19 patients. I agree with the authors that there are limited studies available about brain tissue changes in covid 19 patients, however, a recent search showed that there are at least 4-5 data-based articles that showed the grey matter changes in patients with Covid-19, I would suggest to authors to includes those studies in the review article as the main focus of this article is brain changes and include some figures from those studies if available. Voxel-based brain morphometry: this is the post-processing method and usually runs on the T1 weighted imaging, I think this should include in the analysis as you include the T1 weighted higher resolution imaging to look for grey matter changes.

The authors thank the reviewer for the valuable comment. The works of Douaud et al. and Yiping et al. estimating the grey matter changes in PCC patients were cited in the manuscript (references 9 and 10). The standard brain MRI, which is most likely to be performed for the vast majority of patients as part of a clinical diagnostic procedure, does not include morphometric analysis of isotropic T1-weighted images for several important reasons: (1) the isotropic T1-weighted sequence has long duration and is associated with possible artifacts; (2) radiologists may not have access to specialized software for MRI postprocessing; (3) open tools for DICOM transformation into an isotropic T1-WI (https://doi.org/10.1126/sciadv.add3607) and online voxel-based processing (https://doi.org/10.3389/fninf.2022.862805) do not represent a unified, automatic, and standardized solution for clinical implementation. As the reviewer has requested, we have analyzed publications on voxel-based brain morphometry. In our opinion, the MR findings identified using morphometric analysis are promising and require further study, but they go beyond the scope of this review focused on routine brain MRI findings in PCC.

Reviewer 3 Report

Authors presented an interesting study about the main MRI findings in post COVID19 condition, a disorder that nowadays is widespread. It is important to stress that no alteration is specific per se and all findings are not correlated to the clinical phenotypes. The article is well written and needs just few minor adjustments.

I have few minor suggestion that may increase the quality of the manuscript: 

- the population characteristics (lines 150-158 and 164-167) and general COVID19 features may be listed in a table;

- in the table 3 (PPC MRI characteristics), the number of subjects should be associated to the percentage (%) compared to the single population of study. 

Author Response

Response to Reviewer 3 comments on the manuscript entitled “Routine Brain MRI Findings on Long-Term Effects of COVID-19: A Scoping Review”

Authors presented an interesting study about the main MRI findings in post COVID19 condition, a disorder that nowadays is widespread. It is important to stress that no alteration is specific per se and all findings are not correlated to the clinical phenotypes. The article is well written and needs just few minor adjustments.

We thank the Reviewer for the positive evaluation of our work and useful comments. Our detailed answers are listed below.

  1. The population characteristics (lines 150-158 and 164-167) and general COVID19 features may be listed in a table;

We have added the population characteristics and COVID-19 features to the Table 2.

  1. In the table 3 (PPC MRI characteristics), the number of subjects should be associated to the percentage (%) compared to the single population of study.

We have changed the table numbering in the revised version of the manuscript, and Table 3 is now Table 4. In accordance with the reviewer’s suggestion, we have added the percentages to the total numbers of individual findings.

Reviewer 4 Report

Current study introduces a review on routine brain MRI findings due to COVID-19 Long-Term Effects. 

I appreciate the novelty and the necessity of the review, however, there are some issues I address below:

1. the authors should clearly introduce the reasons why they decided to narrow their review on routine brain MRI findings, especially since they reported that “routine brain MRI protocol has little value for long COVID diagnostics”.

2. lines 194-196: the authors should mention whether the original studies reported that such lesions were attributed exclusively to long Covid or the patients had also other risk factors. 

Author Response

Response to Reviewer 4 comments on the manuscript entitled “Routine Brain MRI Findings on Long-Term Effects of COVID-19: A Scoping Review”

Current study introduces a review on routine brain MRI findings due to COVID-19 Long-Term Effects.

I appreciate the novelty and the necessity of the review, however, there are some issues I address below:

We thank the referee for their kind review. The responses to the comments are provided below:

  1. The authors should clearly introduce the reasons why they decided to narrow their review on routine brain MRI findings, especially since they reported that “routine brain MRI protocol has little value for long COVID diagnostics”.

In response to the comment, we have added the justification for narrowing down the review question to the routine brain MRI (Introduction, first paragraph).

  1. lines 194-196: the authors should mention whether the original studies reported that such lesions were attributed exclusively to long Covid or the patients had also other risk factors.

In the revised version of the manuscript, we have added the necessary details to the description of brain MRI findings (section 3.2).

Round 2

Reviewer 1 Report

I recognize that authors have responded to all my comments appropriately. The manuscripts are improved. This paper can be accepted.